# Phase Transformations and Photocatalytic Activity of Nanostructured Y_2_O_3_/TiO_2_-Y_2_TiO_5_ Ceramic Such as Doped with Carbon Nanotubes

**DOI:** 10.3390/molecules25081943

**Published:** 2020-04-22

**Authors:** Artem L. Kozlovskiy, Inesh Z. Zhumatayeva, Dina Mustahieva, Maxim V. Zdorovets

**Affiliations:** 1Engineering Profile Laboratory, L.N. Gumilyov Eurasian National University, Nur-Sultan 010008, Kazakhstan; inesh.zhumatayeva@gmail.com (I.Z.Z.); mustakhievadi@mail.ru (D.M.); mzdorovets@inp.kz (M.V.Z.); 2Laboratory of Solid State Physics, The Institute of Nuclear Physics, Almaty 050032, Kazakhstan; 3Laboratory of Additive Technologies, Kazakh-Russian International University, Aktobe 030006, Kazakhstan; 4Department of Intelligent Information Technologies, Ural Federal University, 620075 Yekaterinburg, Russia

**Keywords:** carbon nanotubes, doping, titanium oxide, Y_2_TiO_5_ ceramic, phase transformations

## Abstract

This work is devoted to the study of phase transition processes in nanostructured ceramics of the Y_2_O_3_/TiO_2_-Y_2_TiO_5_ type doped with carbon nanotubes as a result of thermal annealing, as well as to the assessment of the prospects of the effect of phase composition on photocatalytic activity. By the method of X-ray phase analysis, it was found that an increase in the annealing temperature leads to the formation of the orthorhombic phase Y_2_TiO_5_, as well as structural ordering. Based on the obtained UV spectra, the band gap was calculated, which varies from 2.9 eV (initial sample) to 2.1 eV (annealed at a temperature of 1000 °C). During photocatalytic tests, it was established that the synthesized nanostructured ceramics Y_2_O_3_/TiO_2_-Y_2_TiO_5_ doped CNTs show a fairly good photocatalytic activity in the range of 60–90% decomposition of methyl orange.

## 1. Introduction

Oxide nanostructured materials and various ceramics based on them have huge practical applications in various industries such as СО_2_ conversion, as catalysts, photocatalysts, and solar cells [1,2,3,4,5]. Among the huge variety of oxide materials or ceramics, titanium dioxide based structures possessing a wide forbidden band of 3.0–3.3 eV, several phase states, high exciton binding energy, good photocatalytic activity, inertness, and biological compatibility have the most promising applications [6,7,8,9]. For example, titanium oxide-based nanostructures have found their application as photocatalysts for the decomposition and subsequent removal of harmful substances by ultraviolet radiation. It should be noted that the photocatalytic activity of titanium oxide is limited by the high recombination rate of electron–positron pairs under the influence of visible light, which leads to a decrease in the efficiency of photocatalysts and their limited use [10,11,12,13]. To eliminate this drawback, as a rule, various methods are used for doping or synthesis of more complex structures with a high density of vacancy positions capable of substantially changing the properties of oxides [14,15,16,17]. Also, in most cases, the use of oxide nanostructured ceramics is associated with external influences, which leads to a decrease in their operational properties due to degradation, lower thermal conductivity, embrittlement, and amorphization [18,19,20,21,22]. In this regard, alloying with rare-earth elements or their oxides is increasingly being used to increase the physicochemical and operational characteristics, to increase the stability of properties and reduce the degradation of materials. One of the ways to increase the stability and efficiency of oxide nanostructured ceramics is to dope them with carbon nanostructures, such as nanotubes, fullerenes, graphene, etc. [23,24,25]. Doping with carbon nanostructures leads to the formation of interfacial heterostructural transitions, an increase in interphase boundaries, as well as conductivity, which has a significant effect on the photocatalytic properties of ceramics [26,27,28,29,30]. 

Based on the foregoing, the main purpose of this work is to study the processes of phase transformations in nanostructured ceramics of the Y_2_O_3_/TiO_2_-Y_2_TiO_5_ type doped with carbon nanotubes as a result of thermal annealing, as well as to evaluate the prospects of the effect of the phase composition on photocatalytic activity.

## 2. Results and Discussion

Figure 1 shows the dynamics of the surface morphology of the nanostructured ceramics Y_2_O_3_/TiO_2_-Y_2_TiO_5_ doped with carbon nanotubes before and after thermal annealing. Figure 1a presents SEM images of the initial nanostructures, which are a mixture of carbon nanotubes coated with spherical dendrites of titanium oxide and yttrium, as evidenced by the results of elemental analysis, presented in Table 1.

In the case of annealing at a temperature of 600 °C, the carbon nanotubes are mixed and partially melted, followed by recrystallization of particles. An increase in the annealing temperature to 800 °C leads to the formation of sphere-like nanoparticles whose average size is from 100–200 nanometers coated with porous inclusions. In this case, a further increase in the annealing temperature to 1000 °C leads to sintering of particles into a porous matrix containing spherical growths, the formation of which is due to the processes of melting of yttrium oxide and the subsequent formation of large particles.

According to X-ray phase analysis, after grinding in the initial state, nanostructured ceramics are a mixture of two phases of the monoclinic phase of yttrium oxide Y_2_O_3_ and the tetragonal phase TiO_2_ characteristic of anatase (see Figure 2). In this case, the presence of amorphous broadening in the range 2*θ* = 18–22° is characteristic of carbon nanostructures, which do not give crystalline reflections. The shape and width of the diffraction peaks indicate a high degree of crystallinity of nanostructured ceramics. An increase in the annealing temperature leads to a change not only in the shape and intensity of diffraction lines, which is caused by thermal annealing of defects and partial recrystallization as a result of an increase in thermal vibrations of atoms, but also in the appearance of new diffraction reflections characteristic of the orthorhombic phase Y_2_TiO_5_, the presence of which and its subsequent increase in contribution with an increase in the annealing temperature and complete dominance in the structure at an annealing temperature of 1000 °C, indicates the processes of phase transformations. The dynamics of phase transformations is shown in Figure 3, the phase contributions were estimated using the Rietveld method, which includes a full-profile analysis of diffraction patterns. Table 2 presents the results of changes in the structural parameters of the studied structures during thermal annealing.

Figure 4 shows the results of measuring the optical absorption spectra and determining the band gap of the synthesized nanostructured ceramics Y_2_O_3_/TiO_2_-Y_2_TiO_5_ doped CNTs depending on the annealing temperature.

The UV spectra of the synthesized samples are characterized by a wide absorption region with peaks in the region from 300 to 400 nm. The presence of peaks at 300–350 nm is due to the sorption and absorbing properties of nanostructured ceramics, it is worth noting that an increase in the annealing temperature, as well as a change in the phase composition with a further predominance of the Y_2_TiO_5_ phase, leads to an increase in the intensity and shift of spectral maxima, which indicates a change in the absorption maxima properties of ceramics. Based on the obtained spectra, the band gap was calculated, which varies from 2.9 eV (initial sample) to 2.1 eV (annealed at a temperature of 1000 °C) depending on the phase composition of ceramics, which is caused by a change in the phase composition and the dominance of the orthorhombic phase Y_2_TiO_5_. In this case, the deviation of the band gap for the initial sample from the values characteristic of titanium oxide is due to deformation processes, as well as substitution processes in the crystal structure due to mechanical grinding processes. A further decrease in the value is due to the processes of phase transformations in the structure of ceramics and a change in structural properties and morphology.

Figure 5 shows the results of changes in the current–voltage characteristics of the studied ceramics depending on the annealing temperature, as well as the dynamics of the change in the value of resistivity.

It can be seen from the presented data that an increase in the annealing temperature leads to a change in the slope of the current–voltage curve, which indicates an increase in current strength and a decrease in the resistance of ceramics. It can be seen that in the case of heat treatment, an increase in the annealing temperature does not lead to a strong decrease in resistance compared with this change for annealed nanostructures at 600 °C in comparison with the initial structures. The decrease in resistance for annealed nanostructured ceramics is due to structural ordering and reduction of defects, as well as phase transformations that occur during annealing of ceramics. The addition of carbon nanotubes with metallic conductivity leads to a decrease in the resistance of structures—which, compared to the initial structures, for which according to scanning electron microscopy the effect of fusion with oxide structures was not observed—decreased by more than 5 times, compared with the fact that the temperature increase. Annealing leads to a decrease in resistance by no more than 1.3–1.5 times.

The photocatalytic ability of the synthesized nanostructured ceramics as a function of annealing temperature was studied by evaluating the decomposition of methyl orange in an aqueous solution with a given initial concentration of 25 mg/L, as a result of exposure to UV radiation with a lamp power of 4.5 mW/cm^2^. Assessment of the degree of degradation was carried out by measuring the UV–vis spectra at different time intervals. Figure 6 shows graphs of changes in photocatalytic degradation.

According to the presented data, it can be seen that the synthesized nanostructured ceramics Y_2_O_3_/TiO_2_-Y_2_TiO_5_ doped CNTs show a fairly good photocatalytic activity. In the case of the initial structures and those annealed at a temperature of 600 °C, the degree of decomposition is much lower, which may be due to the presence of two phases in the structure, as well as a large number of dislocation defects. In the case of single-phase Y_2_TiO_5_ nanostructured ceramics obtained at an annealing temperature of 1000 °C, the degree of decomposition of methyl orange is maximal. Moreover, after 40 min of testing, the degree of decomposition is constant. In the case of repeated tests, the synthesized nanostructured ceramics showed a similar degree of decomposition.

Table 3 presents the results of a comparative analysis of the photocatalytic activity of the synthesized nanostructured ceramics with similar nanostructures based on titanium oxide doped with rare earth elements. 

As can be seen from the data presented in Table 3, the doping of titanium oxide-based nanostructures leads not only to stabilization of its structural features, but also to a significant increase in photocatalytic activity. However, in most cases, this process is time-consuming and energy-consuming, since it is associated with large test time intervals. In our case, obtaining single-phase structures of the Y_2_TiO_5_ doped CNTs type leads not only to a significant acceleration of the reaction rate, but also to the achievement of high photocatalytic degradation rates. It is worth noting that there was no work with a similar phase composition doped with carbon nanostructures. At the same time, there are a large number of studies in which doping with titanium-containing nanostructures by carbon nanostructures leads to a significant increase in photocatalytic activity [38,39,40,41].

## 3. Experimental Part

The preparation of nanostructured ceramics was carried out using the solid-phase synthesis method by mixing salts of yttrium nitrite (YN_3_O_9_·6H_2_O) and titanium oxide (TiO_2_) in an equal stoichiometric ratio of 50:50. The chemical purity of the salts was 99.999%, manufactured by Sigma Aldrich. After intensively mixing the suspended salts in an agate mortar to obtain an isotropic composition, carbon nanotubes (СNT) were added to it at a concentration of 10% of the total mass of the sample. The resulting mixture, after mixing and adding carbon nanotubes, was annealed in an oxygen-containing medium in a muffle furnace at a temperature of 600–1000 °C for 5 h, followed by cooling together with the furnace to reduce the risk of a sharp temperature difference. The choice of temperatures is based on phase transformations due to thermal heating. It should be noted that, as a rule, the processes of phase transformations begin to manifest themselves most vividly at temperatures of 0.2–0.5 T_melting_ (for TiO_2_ T_melting_ = 1843 °C, for Y_2_O_3_ T_melting_ = 2425 °C). Doping with carbon nanotubes leads to an acceleration of phase transformations, which is due to the low melting temperature T_melting_ = 1180 °C. Also, in the case of an increase in the annealing temperature, partial annealing of defects occurring during the synthesis, as well as phase transformations, is observed, as evidenced by changes in the symmetry of the diffraction peaks for the annealed samples. 

Doping with carbon nanotubes induces an increase in the photoactivity of titanium-containing structures, for example, in [38] it was shown that doping of thin TiO_2_ films with CNT leads to a decrease in the band gap and also to an increase in photocatalytic degradation. In this case, for annealed structures at a temperature of 450 °C, an increase in photocatalytic activity is due to the appearance of additional charge transfers through carbon bonds in the structure [38]. A similar picture of the increase in photoactivity upon doping CNT of titanium-containing nanostructures was demonstrated in [39,40,41]. 

The study of the effect of thermal annealing on morphological changes was carried out using the scanning electron microscopy method performed using a Hitachi TM3030 scanning electron microscope (Hitachi Ltd., Chiyoda, Tokyo, Japan). Shooting mode—LEI; current—20 μA; accelerate voltage—2 kV; WD—8 mm. The study of the elemental composition, as well as the mapping of the structures under study in order to determine the equiprobable distribution of elements in the structure, was carried out using the energy dispersive analysis method performed using the EDA Bruker Flash MAN SVE installation (Bruker, Karlsruhe, Germany), at an accelerating voltage of 15 kV.

The study of structural changes, as well as phase transformations as a result of thermal annealing, was carried out using the method of X-ray phase analysis performed on a D8 Advance Eco powder diffractometer (Bruker, Karlsruhe, Germany). Conditions for recording diffractograms: 2*θ* = 15–95°, step 0.01°, Bragg-Brentano geometry, spectrum acquisition time 5 s, X-ray radiation Cu-Kα, λ = 1.54 Å. Structural parameters were determined using the DiffracEva 4.2 program code; the phase composition was determined using the Topas v.4 program code based on the Rietveld method. The volume fraction of the phase contribution was determined using Equation (1)
(1)$$V_{admixture}=\frac{RI_{phase}}{I_{admixture}+RI_{phase}}$$

I_phase_ is the average integral intensity of the main phase of the diffraction line, I_admixture_ is the average integral intensity of the additional phase, R is the structural coefficient equal to 1.45.

The size of crystallites, which was calculated according to the Scherrer equation, Equation (2),
(2)$$\tau =\frac{k\lambda }{\beta \cos\theta }$$
where *k* = 0.9 is the dimensionless particle shape coefficient (Scherrer constant), *λ* = 1.54 Å is the X-ray wavelength, β is the half-width of the reflection at half maximum (FWHM), and *θ* is the diffraction angle (Bragg angle).

The study of the absorption spectra of the studied nanostructured ceramics was obtained using the UV spectroscopy method using a Jena Specord-250 BU analytical spectrophotometer equipped with an integrating sphere. BaSO_4_ was used as the standard. The resolution is 1 nm, and the scanning speed was 20 nm/s. The spectral range is 190–1100 nm.

## 4. Conclusions 

The work is devoted to the study of phase transition processes in nanostructured ceramics of the Y_2_O_3_/TiO_2_-Y_2_TiO_5_ type doped with carbon nanotubes as a result of thermal annealing, as well as to the assessment of the promising effect of the phase composition on photocatalytic activity. The preparation of nanostructured ceramics was carried out using the solid-phase synthesis method by mixing the salts of the starting components, followed by the addition of carbon nanotubes and thermal annealing in the temperature range 600–1000 °C. Based on the obtained UV spectra, the band gap was calculated, which varies from 2.9 eV (initial sample) to 2.1 eV (annealed at a temperature of 1000 °C). It has been established that doping with carbon nanotubes with metallic conductivity, as well as subsequent thermal annealing initiating phase transition processes, leads to a decrease in the resistance of structures. During photocatalytic tests, it was established that the synthesized nanostructured ceramics Y_2_O_3_/TiO_2_-Y_2_TiO_5_ doped CNTs show fairly good photocatalytic activity in the range of 60–90% decomposition of methyl orange.

## Figures and Tables

**Figure 1 molecules-25-01943-f001:**
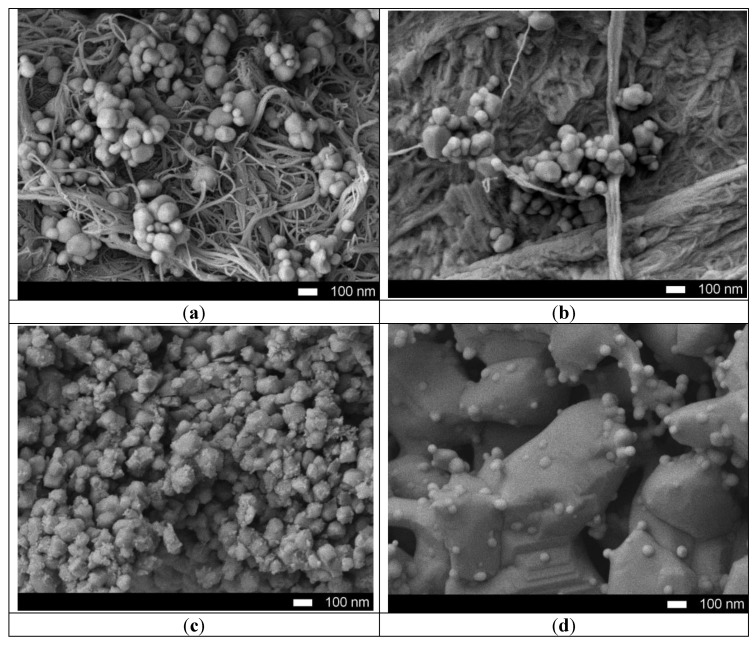
SEM images of synthesized nanostructured ceramics Y_2_O_3_/TiO_2_-Y_2_TiO_5_ doped CNTs: (**a**) 25 °C; (**b**) 600 °C; (**c**) 800 °C; (**d**) 1000 °C.

**Figure 2 molecules-25-01943-f002:**
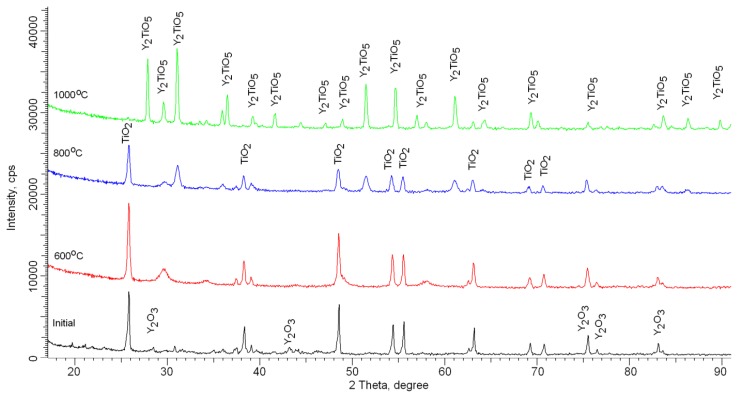
Dynamics of X-ray diffraction patterns of nanostructured ceramics Y_2_O_3_/TiO_2_-Y_2_TiO_5_ doped CNTs.

**Figure 3 molecules-25-01943-f003:**
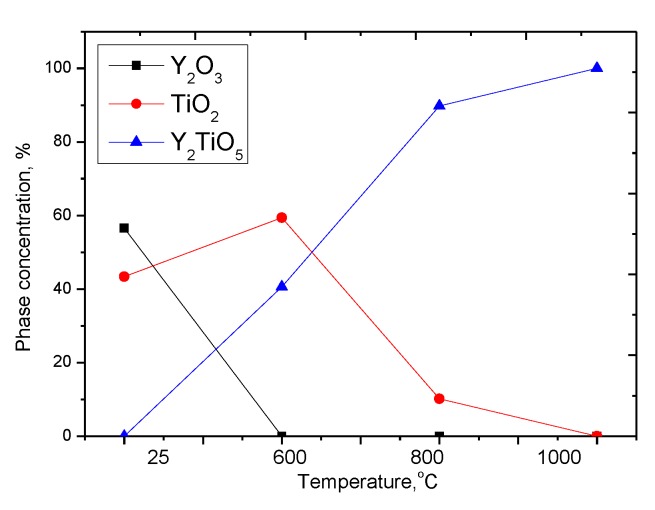
Dynamics of phase transformations of nanostructured ceramics Y_2_O_3_/TiO_2_-Y_2_TiO_5_ doped CNTs.

**Figure 4 molecules-25-01943-f004:**
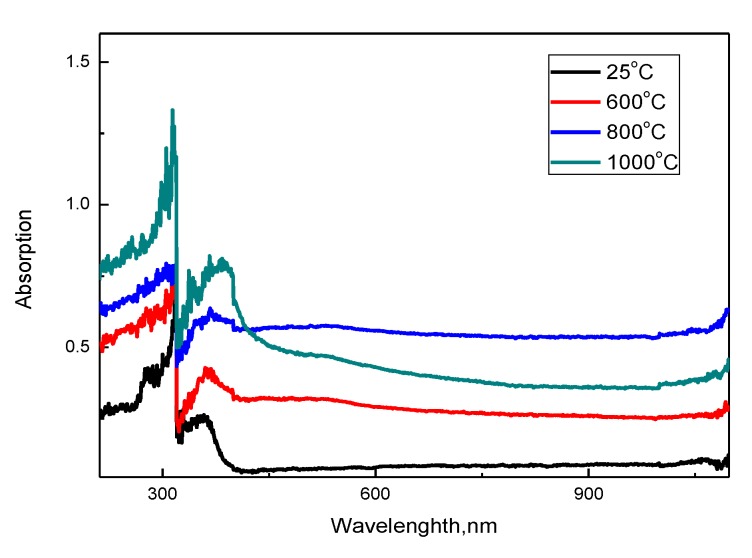
Results of UV spectroscopy of the studied nanostructured ceramics Y_2_O_3_/TiO_2_-Y_2_TiO_5_ doped CNTs.

**Figure 5 molecules-25-01943-f005:**
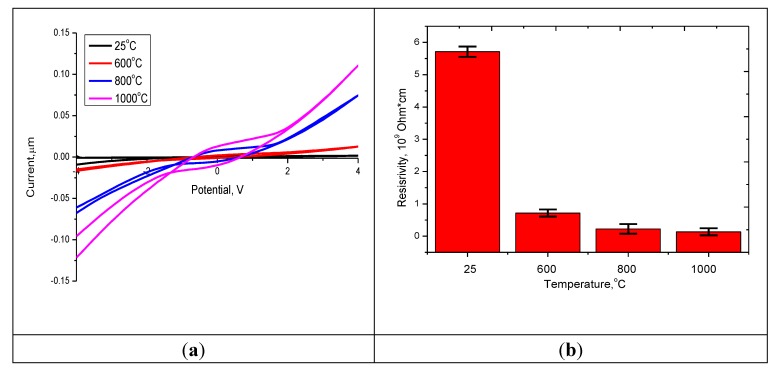
(**a**) I–V characteristics of the studied nanostructured ceramics Y_2_O_3_/TiO_2_-Y_2_TiO_5_ doped CNTs. (**b**) The dynamics of changes in the resistance of the studied nanostructured ceramics.

**Figure 6 molecules-25-01943-f006:**
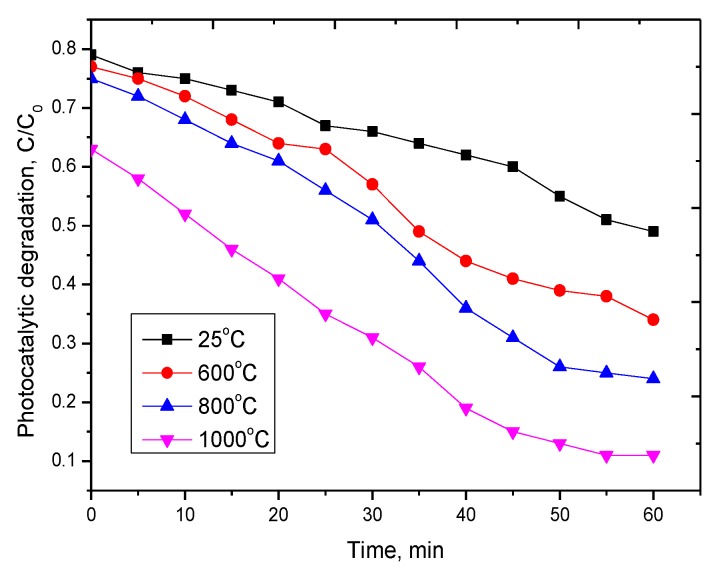
Dynamics of changes in the photocatalytic degradation of methyl orange on nanostructures, where С_0_ is the concentration of methyl orange in an aqueous solution of 25 mg/L, C is the concentration of methyl orange after irradiation with UV radiation.

**Table 1 molecules-25-01943-t001:** Elemental analysis of nanostructures Y_2_O_3_/TiO_2_-Y_2_TiO_5_ doped CNTs.

Annealing Temperature	Elemental Content, at %			
	Oxygen (O)	Titanium (Ti)	Yttrium (Y)	Carbon (С)
Initial sample	52 ± 6 *	15 ± 2	14 ± 1	19± 2
600 °C	52 ± 5	17 ± 1	14 ± 2	17 ± 2
800 °C	66 ± 6	18 ± 1	6 ± 1	10± 1
1000 °C	60 ± 7	23 ± 1	5 ± 1	12 ± 2

* Measurement errors were determined by evaluating 10 spectra from different sections of the test sample and then determining the average value and standard deviation.

**Table 2 molecules-25-01943-t002:** Data of structural parameters.

Sample	Phase, Space Group	Lattice Parameter, Å	Crystalline Size, nm
Initial	Y_2_O_3_—Monoclinic C2/m(12)	a = 14.150 ± 0.012, b = 3.521 ± 0.009 *, c = 8.721 ± 0.005, beta = 99.91° V = 428.01 Å^3^	45 ± 2 **
	TiO_2_—Tetragonal I41/amd(141)	a = 3.750 ± 0.014, c = 9.421 ± 0.009 V = 132.45 Å^3^	41 ± 4
600 °C	Y_2_TiO_5_—Orthorhombic Pnam(62)	a = 10.311 ± 0.009, b = 11.144 ± 0.005, c = 3.681 ± 0.011, V = 422.95 Å^3^	10 ± 2
	TiO_2_—Tetragonal I41/amd(141)	a = 3.712 ± 0.013, c = 9.515 ± 0.013 V = 131.11 Å^3^	32 ± 2
800 °C	Y_2_TiO_5_—Orthorhombic Pnam(62)	a = 10.252 ± 0.011, b = 11.080 ± 0.009, c = 3.656 ± 0.006, V = 415.29 Å^3^	20 ± 3
	TiO_2_—Tetragonal I41/amd(141)	a = 3.701 ± 0.009, c = 9.487 ± 0.004 V = 129.96 Å^3^	30 ± 2
1000 °C	Y_2_TiO_5_—Orthorhombic Pnam(62)	a = 10.150 ± 0.011, b = 11.057 ± 0.009, c = 3.632 ± 0.007, V = 407.59 Å^3^	40 ± 3

* The crystal lattice parameter was determined by comparative analysis of the positions of the diffraction peaks and comparing them with the positions of the reference card values from the PDF-2 database. The measurement error was also determined by calculating standard deviations using this program code. The parameters were refined by the main diffraction peaks characteristic of each phase; ** The crystallite size was determined using the Scherrer formula, by analyzing all diffraction peaks, determining the average value and standard deviation.

**Table 3 molecules-25-01943-t003:** Comparative analysis data.

Structure Type	Reaction Type	Summary of the Results	References
Yttrium-doped TiO_2_ nanosheet-array films	Photocatalytic degradation of MO aqueous solution under the simulated solar light irradiation	It was established that Y–TiO_2_ films with a dopant content of 2.5 and 5 wt % showed the highest photocatalytic activity with a decrease in the dye concentration of more than 80% after 6 h of the reaction.	[31]
HPW-Y-TiO_2_ composites	Degradation kinetics of methyl orange under UV ligh	Dependences between the concentration of dopant and various conditions for conducting photocatalytic reactions are established. It is also shown that doping leads to a sharp increase in the rate of photocatalytic degradation.	[32]
Y^3+^-doped TiO_2_ nanoparticles	Degradation kinetics of methyl orange under UV light	It was found that doping with yttrium (1.5 mol %) And subsequent thermal annealing lead to an increase in the photodegradation rate and degree of decomposition to 99.8% under the influence of UV radiation for very short time periods.	[33]
TiO_2_ and TiO_2_/Y2O3 nanoparticles were prepared by sol-gel method	Degradation of methylene blue under UV and visible light illumination	Structures in which the concentration of doped Y_2_O_3_ was 0.8–1.0 wt %, as well as a mixture of titanium oxide phases: rutile and anatase, have the highest photocatalytic activity. It was shown that the presence of multiphase in the structure plays a double role in the photocatalytic activity of structures.	[34]
Yttrium-doped TiO_2_ microspheres	Photocatalytic activity was evaluated by measuring the degradation rate of methyl orange (MO) solution under visible irradiation	It has been shown that structures in which the doping concentration of yttrium is not more than 1–1.5%, the photodegradation value is 0.3–0.4, and the photodegradation time is more than 300 min have the highest photoactivity.	[35]
Rare earth doped TiO_2_ nanoparticles	Photocatalytic activity was evaluated by the photocatalytic decomposition of Orange II dye in an aqueous solution	It was established that doping with rare-earth elements (0.5–1 wt %) leads to a significant increase in photoactivity, which is associated with the separation of charge carriers. In this case, the structures doped with Nd showed the highest photoactivity.	[36]
La doped TiO_2_	Photocatalytic phenol decomposition	It was shown that for annealed structures above 500–600 °C, a decrease in the photocatalytic degradation of phenol is observed, which is 0.8–0.83 for structures obtained by annealing at 500–600°C and 0.85–0.87 for structures obtained at 800 °C.	[37]
Nanostructured ceramics Y_2_O_3_/TiO_2_-Y_2_TiO_5_ doped CNTs	Decomposition of methyl orange in an aqueous solution with a given initial concentration of 25 mg/L	It was found that for annealed structures, the degree of decomposition of methyl orange is much higher than for the initial structures. The decrease in С/С_0_ concentration for annealed structures is in the range of 0.1–0.4.	This work

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
