# Peer review of "Phase Transformations and Photocatalytic Activity of Nanostructured Y2O3/TiO2-Y2TiO5 Ceramic Such as Doped with Carbon Nanotubes"

_molecules, 2020, doi:10.3390/molecules25081943_

Round 1

Reviewer 1 Report

The manuscript entitled "Phase transformations and photocatalytic activity of nanostructured Y2O3/TiO2–Y2TiO5 ceramic such as doped with carbon nanotubes" with (manuscript ID: molecules–779760) provides experimental observations for ceramics doped with CNT. Both the materials and the research topic is of high importance in the field of catalysis and to the scientific community. The manuscript is appropriate for publication in the “Molecules”. In the present form, the manuscript is not suitable for publication as it lacks important information– the details of which are provided in the following.

  1. The authors present the results such as phase transformation, resistivity, and photocatalytic degradation activity for the CNT-doped ceramic. However, a comparison of the above (marked as bold letters) with respect to the undoped ceramic case is missing.
  2. The reason for considering the three specific annealing temperatures should be explained.

After the inclusion of the above comments, I would recommend the manuscript to be accepted for publication in the journal.

Author Response

1. The authors are very grateful for this remark.

The main goal of this work was to consider the changes in the properties of the selected Y2O3/TiO2-Y2TiO5 doped CNTs system.

In the near future, it is planned to complete the study of the effect of doping not only with carbon nanostructures, but also with other elements of the selected system of titanium-containing ceramics, within the framework of which the results of a comparative analysis of various synthesis conditions on the properties of ceramics will be presented.

Unfortunately, at the moment, it is impossible to finish these works due to various reasons, therefore, the team of authors asks to treat this fact with understanding.

However, this work presents the results of a comparative analysis of photocatalytic activity with other works currently available in search databases that consider the results of doping.

2. The choice of temperatures is based on phase transformations due to thermal heating. It should be noted that, as a rule, the processes of phase transformations begin to manifest themselves most vividly at temperatures of 0.2-0.5 Tmelting (for TiO2 Tmelting = 1843°Ð¡, for Y2O3 Tmelting=2425°Ð¡). Doping with carbon nanotubes leads to an acceleration of phase transformations, which is due to the low melting temperature Tmelting=1180°Ð¡. Also, in the case of an increase in the annealing temperature, partial annealing of defects occurring during the synthesis, as well as phase transformations, is observed, as evidenced by changes in the symmetry of the diffraction peaks for the annealed samples.  

Reviewer 2 Report

The manuscript reports on the preparation of ittria/titania nanoparticles, doped with carbon nanotubes. The manuscript is well organized but some major issues must be solved before evaluationg the possibility of publication.

Figure 1 (SEM) needs a restyle: the scale can not be seen. A comment on the effect of temperature on nanotubes and Yittria and Titania alone from the literature is mandatory to see if the annealing of the mixture is similar or different to the expect effect of annealing on the two pure ingredients. Line 63-69. Please specify the model and main measurement conditions for SEM, XRD and UV analyses. Elemental analysis appears in table 1 caption. It must detailed also in the experimental section: is this EDS associated to SEM?. The details of Rietveld analysis must be reported in the experimental section too. Table 1: error on the first decimal digit is non sense: it is larger for sure. Table 2: The error on the first decimal digit is non sense, it is larger for sure. Again, the method used to obtain the “crystalline size” must be detailed, as well how the error associated to the measurement was carried out. Errors on lattice parameters must be reported in Table 2. The results (especially lines 157-164 and the conclusion) must be discussed in comparison with what already known in the literature. If not, it is impossible appreciating the novelties of the present work.

A final concern is related to the journal type: the manuscript does not refers to “molecules” but more to materials and catalysts. I wonder if this article might be more suitable for Catalysts or Materials journals. But this consideration is left to the editor. After these major notes (and the minor here below detailed) are fixed the manuscript can be re-evaluated for publication.

Minor notes

  • 1) Line 55 “99.999 %” This purity is an analytical or pharmaceutical grade rather unusual in materials preparation? Is this purity correct? Is it possible transferring in the real world the production of materials using so high purity (and high cost I guess)? Is this high purity mandatory? Standard reactant 99.7% purity materials could work instead?
  • 2) Line 73: “after mixing, the initial structures are a mixture” is a tautology, please correct
  • 3) Line 94 Use “diffraction peaks” instead of “diffraction lines”
  • 4) Line 136: “the angle of inclination” I guess authors refer to the “slope”
  • 5) Line 145 “increase Annealing” is a “dot” lacking?

Author Response

1. The choice of temperatures is based on phase transformations due to thermal heating. It should be noted that, as a rule, the processes of phase transformations begin to manifest themselves most vividly at temperatures of 0.2-0.5 Tmelting (for TiO2 Tmelting = 1843°Ð¡, for Y2O3 Tmelting=2425°Ð¡). Doping with carbon nanotubes leads to an acceleration of phase transformations, which is due to the low melting temperature Tmelting=1180°Ð¡. Also, in the case of an increase in the annealing temperature, partial annealing of defects occurring during the synthesis, as well as phase transformations, is observed, as evidenced by changes in the symmetry of the diffraction peaks for the annealed samples.

Doping with carbon nanotubes is due to an increase in the photoactivity of titanium-containing structures, for example, in [31] it was shown that doping of thin TiO2 films with CNT leads to a decrease in the band gap and also to an increase in photocatalytic degradation. In this case, for annealed structures at a temperature of 450°Ð¡, an increase in photocatalytic activity is due to the appearance of additional charge transfers through carbon bonds in the structure [31]. A similar picture of the increase in photoactivity upon doping CNT of titanium-containing nanostructures was demonstrated in [32-34].

a)

b)

c)

d)

Figure 1. SEM images of synthesized nanostructured ceramics Y2O3/TiO2-Y2TiO5 doped CNTs: a) 25°Ð¡; b) 600°Ð¡; c) 800°Ð¡;  d) 1000°Ð¡.

The study of the effect of thermal annealing on morphological changes was carried out using the scanning electron microscopy method performed using a “Hitachi TM3030” scanning electron microscope (Hitachi Ltd., Chiyoda, Tokyo, Japan). Shooting mode - LEI, Current – 20 μA, Accelerate voltage – 2 kV, WD= 8 mm. The study of the elemental composition, as well as the mapping of the structures under study in order to determine the equiprobable distribution of elements in the structure, was carried out using the energy dispersive analysis method performed using the EDA Bruker Flash MAN SVE installation (Bruker, Karlsruhe, Germany), at an accelerating voltage of 15 kV.

The study of structural changes, as well as phase transformations as a result of thermal annealing, was carried out using the method of x-ray phase analysis performed on a D8 Advance Eco powder diffractometer (Bruker, Karlsruhe, Germany). Conditions for recording diffractograms: 2θ = 15–95°, step 0.01°, Bragg-Brentano geometry, spectrum acquisition time 5 s, X-ray radiation Cu-Kα, λ = 1.54 Å. Structural parameters were determined using the DiffracEva 4.2 program code; the phase composition was determined using the Topas v.4 program code based on the Rietveld method. The volume fraction of the phase contribution was determined using equation (1):

(1)

Iphase is the average integral intensity of the main phase of the diffraction line, Iadmixture is the average integral intensity of the additional phase, R is the structural coefficient equal to 1.45.

The size of crystallites, which was calculated according to the Scherrer equation (2):

(2)

where k = 0,9 is the dimensionless particle shape coefficient (Scherrer constant), λ=1,54Å is the X-ray wavelength, β is the half-width of the reflection at half maximum (FWHM), and θ is the diffraction angle (Bragg angle).

The study of the absorption spectra of the studied nanostructured ceramics was obtained using the UV spectroscopy method using a Jena Specord-250 BU analytical spectrophotometer equipped with an integrating sphere. BaSO4 was used as the standard. The resolution is 1 nm, and the scanning speed was 20 nm/s. The spectral range is from 190 to 1100 nm.

Corrected.

Table 1.

* Measurement errors were determined by evaluating ten spectra from different sections of the test sample and then determining the average value and standard deviation.

Table 2.

*The crystal lattice parameter was determined by comparative analysis of the positions of the diffraction peaks and comparing them with the positions of the reference card values from the PDF-2 database, in the software package for decoding X-ray diffraction patterns DiffracEVA v.4.2, (Bruker). The measurement error was also determined by calculating standard deviations using this program code. The parameters were refined by the main diffraction peaks characteristic of each phase.

**The crystallite size was determined using the Scherrer formula, by analyzing all diffraction peaks, determining the average value and standard deviation.

Table 3 presents the results of a comparative analysis of the photocatalytic activity of the synthesized nanostructured ceramics with similar nanostructures based on titanium oxide doped with rare earth elements.

Table 3. Comparative analysis data.

Structure type

Reaction type

A brief description of the results

References

Yttrium-doped TiO2 nanosheet-array films

Photocatalytic degradation of MO aqueous solution under the simulated solar light irradiation

It was established that Y–TiO2 films with a dopant content of 2.5 and 5 wt. % showed the highest photocatalytic activity with a decrease in the dye concentration of more than 80% after 6 hours of the reaction.

[35]

HPW-Y-TiO2 composites

The degradation kinetics of methyl orange under UV ligh

The dependences between the concentration of dopant and various conditions for conducting photocatalytic reactions are established. It is also shown that doping leads to a sharp increase in the rate of photocatalytic degradation.

[36]

Y3+-doped TiO2 nanoparticles

The degradation kinetics of methyl orange under UV ligh

It was found that doping with yttrium (1.5 mol.%) And subsequent thermal annealing lead to an increase in the photodegradation rate and degree of decomposition to 99.8% under the influence of UV radiation for very short time periods.

[37]

TiO2 and TiO2/Y2O3 nanoparticles were prepared by sol-gel method

Degradation of methylene blue under UV and visible light illumination

Structures in which the concentration of doped Y2O3 was 0.8-1.0 wt.%, as well as a mixture of titanium oxide phases: rutile and anatase, have the highest photocatalytic activity. It was shown that the presence of multiphase in the structure plays a double role in the photocatalytic activity of structures.

[38]

Yttrium-doped TiO2 microspheres

The photocatalytic activity were evaluated by measuring the degradation rate of methyl orange (MO) solution under visible irradiation

It has been shown that structures in which the doping concentration of yttrium is not more than 1-1.5%, the photodegradation value is 0.3-0.4, and the photodegradation time is more than 300 minutes have the highest photoactivity.

[39]

Rare earth doped TiO2 nanoparticles

Photocatalytic activity was evaluated by the photocatalytic decomposition of Orange II dye in an aqueous solution.

It was established that doping with rare-earth elements (0.5-1 wt.%) leads to a significant increase in photoactivity, which is associated with the separation of charge carriers. In this case, the structures doped with Nd showed the highest photoactivity.

[40]

La doped TiO2

Photocatalytic phenol decomposition

It was shown that for annealed structures above 500-600°C, a decrease in the photocatalytic degradation of phenol is observed, which is 0.8-0.83 for structures obtained by annealing at 500-600°C and 0.85-0.87 for structures obtained at 800°C.

 [41]

Nanostructured ceramics Y2O3/TiO2-Y2TiO5 doped CNTs

The decomposition of methyl orange in an aqueous solution with a given initial concentration of 25 mg/l

It was found that for annealed structures, the degree of decomposition of methyl orange is much higher than for the initial structures. The decrease in С/С0 concentration for annealed structures is in the range 0.1-0.4.

This work

As can be seen from the data presented in table 3, the doping of titanium oxide-based nanostructures leads not only to stabilization of its structural features, but also to a significant increase in photocatalytic activity. However, in most cases, this process is time-consuming and energy-consuming, since it is associated with large test time intervals. In our case, obtaining single-phase structures of the Y2TiO5 doped CNTs type leads not only to a significant acceleration of the reaction rate, but also to the achievement of high photocatalytic degradation rates. It is worth noting that there was no work with a similar phase composition doped with carbon nanostructures when conducting searches for keywords. At the same time, there are a large number of studies in which doping with titanium-containing nanostructures by carbon nanostructures leads to a significant increase in photocatalytic activity [31-34].

The authors are grateful to the referee for this comment.

These materials had an analytical purity of 99.999%. The choice in favor of these components was made in view of their presence in the laboratory. You can also replace these reagents with reagents with a standard purity of 99.7-99.9%.

Figure 1a presents SEM images of the initial nanostructures, which are a mixture of carbon nanotubes coated with spherical dendrites of titanium oxide and yttrium, as evidenced by the results of elemental analysis, presented in table 1.

The authors are grateful to the referee for this comment.

All inaccuracies are resolved.

Round 2

Reviewer 2 Report

All the proposed suggestions/requests were fulfilled and the manuscript could be accepted provided that some minor issues are fixed:

  • Line 70: the sentence “Doping with carbon nanotubes is due to an increase in the photoactivity” should be rewritten. “induces” instead of is due to”?
  • In the first note of Table 2 the software DiffractEVA is indicated. In the second note about sherrer analysis the used method/software is not cite. I suggest to remove the note to EVA since is already written in exzperimental section. Or to insert the same note for sherrer analysis for sake of consistency, but I prefer the first option
  • Line 170 still “in the slope of inclination of the current – voltage characteristic”. I suggest a mucg simpler to read “in the slope of the current – voltage curve”
  • Line 212 please remove “when conducting searches for keywords”

Author Response

Line 70: the sentence “Doping with carbon nanotubes is due to an increase in the photoactivity” should be rewritten. “induces” instead of is due to”?

Doping with carbon nanotubes induces an increase in the photoactivity of titanium-containing structures, for example, in [31] it was shown that doping of thin TiO2 films with CNT leads to a decrease in the band gap and also to an increase in photocatalytic degradation.

In the first note of Table 2 the software DiffractEVA is indicated. In the second note about sherrer analysis the used method/software is not cite. I suggest to remove the note to EVA since is already written in exzperimental section. Or to insert the same note for sherrer analysis for sake of consistency, but I prefer the first option.

*The crystal lattice parameter was determined by comparative analysis of the positions of the diffraction peaks and comparing them with the positions of the reference card values from the PDF-2 database. The measurement error was also determined by calculating standard deviations using this program code. The parameters were refined by the main diffraction peaks characteristic of each phase.

**The crystallite size was determined using the Scherrer formula, by analyzing all diffraction peaks, determining the average value and standard deviation.

Line 170 still “in the slope of inclination of the current – voltage characteristic”. I suggest a mucg simpler to read “in the slope of the current – voltage curve”

It can be seen from the presented data that an increase in the annealing temperature leads to a change in the slope of the current – voltage curve, which indicates an increase in current strength and a decrease in the resistance of ceramics.

Line 212 please remove “when conducting searches for keywords”

In our case, obtaining single-phase structures of the Y2TiO5 doped CNTs type leads not only to a significant acceleration of the reaction rate, but also to the achievement of high photocatalytic degradation rates. It is worth noting that there was no work with a similar phase composition doped with carbon nanostructures.
